# Application of Antioxidants as an Alternative Improving of Shelf Life in Foods

César Leyva-Porras [1], Manuel Román-Aguirre [2], Pedro Cruz-Alcantar [3], José T. Pérez-Urizar [4] and María Zenaida Saavedra-Leos [4,*]

1  Centro de Investigación en Materiales Avanzados S.C. (CIMAV), Miguel de Cervantes 120, Complejo Industrial Chihuahua, Chihuahua 31136, Mexico; cesar.leyva@cimav.edu.mx
2  Doctorado Institucional en Ingeniería y Ciencia de los Materiales, Universidad Autónoma de San Luis Potosí, San Luis Potosí 78210, Mexico; iq_manuelroman@yahoo.com
3  Coordinación Académica Región Altiplano, Universidad Autónoma de San Luis Potosí, Carretera Cedral Km. 5+600 Ejido San José de las Trojes, Matehuala 78700, Mexico; pedro.cruz@uaslp.mx
4  Facultad de Ciencias Químicas, Universidad Autónoma de San Luis Potosí, San Luis Potosí 78210, Mexico; jpurizar@uaslp.mx
*  Correspondence: zenaida.saavedra@uaslp.mx; Tel.: +52-(488)-125-0150

**Abstract:** Oxidation is the main problem in preserving food products during storage. A relatively novel strategy is the use of antioxidant-enriched edible films. Antioxidants hinder reactive oxygen species, which mainly affect fats and proteins in food. At present, these films have been improved by the addition of micro- and nanoliposomes coated with carbohydrate polymers, which are not hazardous for human health and can be ingested without risk. The liposomes are loaded with different antioxidants, and their effects are observed as a longer storage time of the food product. The synergy of these methodologies and advances can lead to the displacement of the protective packaging used currently, which would result in food products with functional properties added by the films, an increase in shelf life, and an improvement to the environment by reducing the amount of waste.

**Keywords:** polysaccharides; antioxidants; loaded liposomes; edible films; food conservation

## 1. Introduction

Nowadays, one of the biggest challenges faced in the food industry is the rapid spoilage of foods [1]. Evidently, this problem results in a huge loss of money and large amounts of food in the world. Spoilage may be caused by three mechanisms named as the growth of microorganisms, oxidation, and enzymatic self-decomposition. The first mechanisms refers to the naturally occurring or externally added microorganisms, their growth and proliferation within the food [2]. Oxidation is the result of the action of reactive oxygen species (ROS) such as oxygen-containing free radicals ($^-O$), hydroxyl radicals ($^-OH$), and superoxide anions ($^-O_2$) that may react with lipids and proteins, leading to an oxidative state [2–4]. Enzymatic decomposition is caused by the enzymes present in plant or animal cells that break down the fats and proteins of the food. An alternative to counter these mechanisms is the addition of substances that may inhibit the oxidative spoilage before vital molecules may be damaged. These substances are known as antioxidants, and are capable of hindering the effects of the ROS, either avoiding the chain chemical reactions or neutralizing the ROS [5]. There are natural and synthetic antioxidants, but the latter have been related to possible harm to human health [6,7]. Naturally occurring antioxidants are classified according to the chemical structure in carotenoids, vitamins, polyphenols, quinones, and minerals. Unfortunately, exposure to atmospheric air causes natural antioxidants to break down quickly, losing their properties and limiting their antioxidant activity [8–11]. One strategy to prolong their effect is through microencapsulation with

carrying agents. These agents are based on carbohydrate polymers, which in turn are made up of polysaccharide chains such as glucose and fructose [12–14].

Several authors have discussed different methodologies for obtaining edible films such as wet (casting) and dry (extrusion) processes, and classified the films into edible coatings (applied directly to the food product), and preformed films (which wrap the product) [15, 16]. They addressed the challenges faced by food manufacturers in relation to the loss of quality during storage, the subsequent deterioration, and the increase in waste. Although the packaging is an alternative to protect the quality of food, improve its shelf life, delaying microbial deterioration and providing barrier properties against moisture and gases, typical polymers used for this purpose may represent a risk for human health, as migration of molecules such as additives into the food from the polymeric matrix may occur. Another disadvantage is the environmental mark left by the plastic coating once it is disposed as solid waste. In consequence, this methodology is not environmentally friendly. Due to the above, several efforts have been made in order to substitute the traditional polymer packing by more environmental friendly materials such as biopolymers [17–19]. In this sense, a potential alternative to solve this problem with a low impact on the environment is the use of edible films made from natural polymers such as polysaccharides, proteins and fats [20–25].

The aim of this work is to present the application of antioxidants in foods as an alternative to increase the shelf life of the product. The contribution of the work lies in the use of polysaccharides as carrier agents in the development of antioxidants loaded liposomes and their incorporation in edible films. This work presents: (i) the oxidation processes of lipids and proteins, which are the main components of the foods, and their chemical reaction mechanisms; (ii) the reducing function of naturally occurring antioxidants; (iii) the use of nanometric lipids and liposomes as vehicles in the conservation of antioxidants, and (iv) the application of the antioxidant-loaded liposomes as a protective barrier in edible films. Although the bibliographic search included articles published more than 35 years ago, most of the works presented herein were published very recently, i.e., during the last 5 years.

## 2. Food Oxidation Processes

As mentioned, food can deteriorate due to several factors such as oxidation, which may be caused by the effects of UV light, ambient or cooking temperature and agrochemical residues. Proteins, lipids and sugars contained in food can suffer deterioration due to oxidation. Apart from the effect on the appearance and the loss of nutritional value, foods that have undergone oxidative processes can be harmful to health [12,26]. One of the main problems of the deterioration of beef, fish, and their derivatives, is the oxidation of lipids and proteins, which is produced by the uncontrolled generation of free radicals and reactive species [27]. These changes cause a deterioration in quality in terms of taste, color, texture and nutritional value. Additionally, it is very important to know the oxidation mechanism of fats and proteins, and their relationship with the majority component of foods, for example, proteins, lipids, and carbohydrates, in order to predict the oxidation rate and shelf life of the product.

For lipids or fats, Jackson, et al. [28] conducted a study on the effect of oxidized lipids on health, specifically to set a relationship between the intake of oxidized lipids and the risk of developing chronic diseases. They suggested that the processes that include antioxidants to preserve food must be studied to confirm that they promote health and reduce the risk of developing diseases. Maldonado-Pereira et al. [29] studied the oxidation of cholesterol and its effect on food toxicity. They reported that there is abundant scientific evidence regarding the correlation between the consumption of cholesterol oxidation products and the risk of developing cancer and degenerative neurological diseases. Kato, et al. [30] published a methodology based on chromatography and mass spectroscopy in the quantification of the degree of oxidation in lipids, with greater sensitivity than that measured with the traditional method of the peroxide value (POV). They found that the

phootoxidation of lipids produces peroxides on unsaturated carbons different from those obtained by thermo-oxidation. Waraho et al. [31] studied the reaction mechanisms in the oxidation of lipids in emulsion foods, such as dairy products. They found that some sugars such as hexose and pentose might act as pro-oxidants of lipids in emulsions. Additionally, because of the different characteristics of surfactants and emulsifiers, they suggested the use of antioxidants to reduce the possible oxidation and rancidity of emulsified products. Figures 1 and 2 describe the oxidation reaction mechanism by hydroxyl radicals of an unsaturated lipid (L). The hydroxyl radical causes the homolytic cleavage of the hydrogen bond of the $sp^2$ carbon, giving rise to a lipid radical •L. The lipid radical reacts with an $O^2$ molecule to form a lipid-peroxide radical (LOO•). This reactive species can propagate lipid oxidation by taking hydrogen from another molecule and forming the lipid-hydroperoxide species (LOOH) and a new lipid radical susceptible to oxidation. It can also happen that two lipid radicals react with each other, leading to lipid crosslinking (Figure 2) [32].

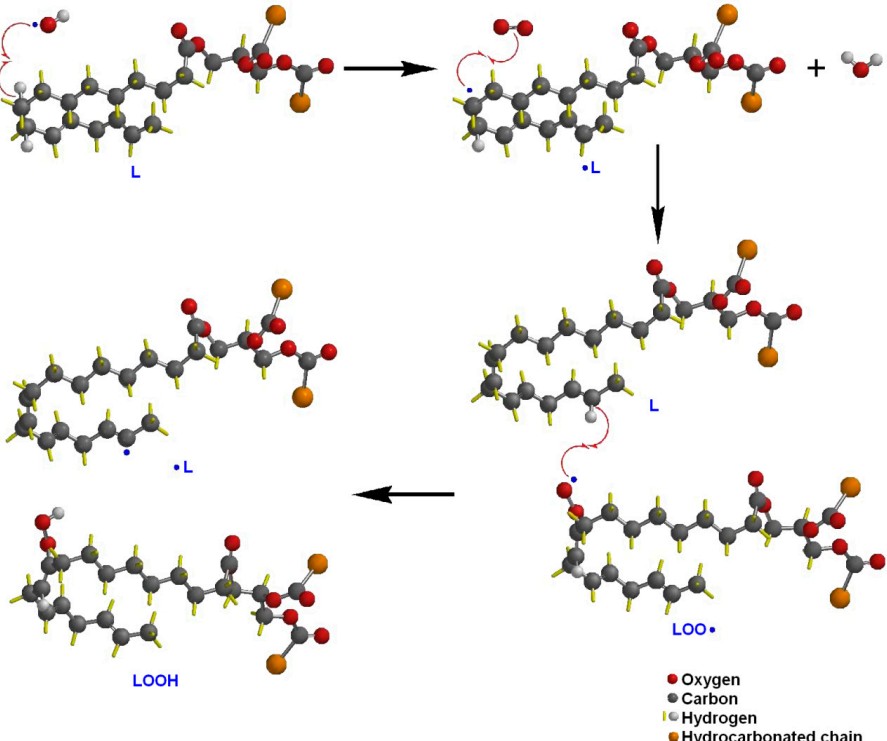

**Figure 1.** Oxidation reaction mechanism by hydroxyl radicals of an unsaturated lipid (L).

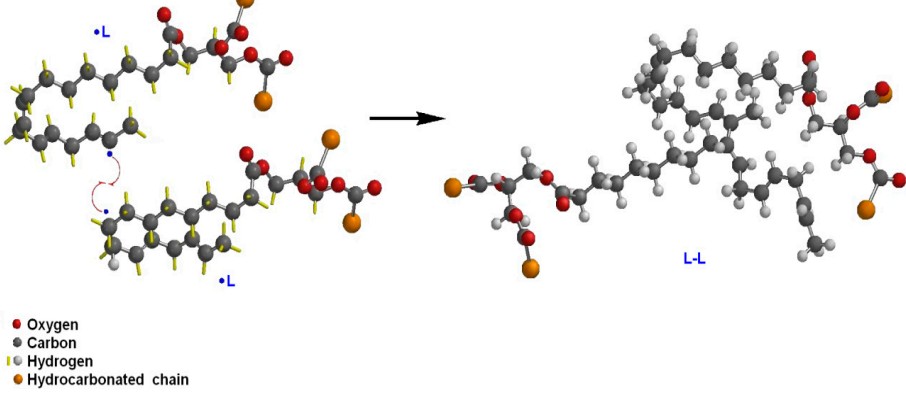

**Figure 2.** Lipid crosslinking mechanism from the reaction of two lipid radicals.



For proteins, Estévez and Luna [33] studied the effects on health due to the oxidation of proteins in food. They mentioned that some proteins or oxidized amino acids consumed in food can induce cellular malfunction, both due to a deterioration in self-regulation of the organism (homeostasis) and programmed cell death (apoptosis). Hellwig [34] carried out an extensive review on the oxidation mechanisms of proteins in food. Among the main sources of oxidizing species, he included ultraviolet light and transition metals such as iron. In addition, some polyphenols that inhibit lipid oxidation can act as pro-oxidants in proteins, but that this effect can be controlled with low concentrations of antioxidants. The oxidation mechanism in proteins is described in Figure 3. A reactive oxygen species can react with an amino acid chain (P) by reacting with hydrogen from one of the carbons, thus forming a free radical in the chain (●P). This radical reacts with an oxygen molecule forming a protein-peroxide radical (POO●) which in turn can react with hydrogen from another chain of amino acids creating a hydroperoxide (POOH) and a new free radical. The reaction between two protein radicals leads to the cross-linking of amino acid chains as shown in Figure 4 [35].

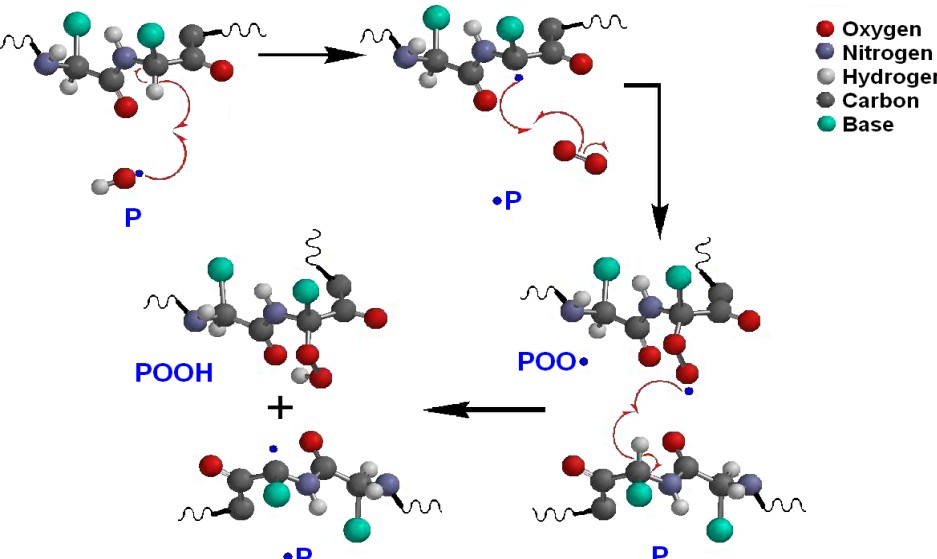

**Figure 3.** Oxidation mechanism in proteins.

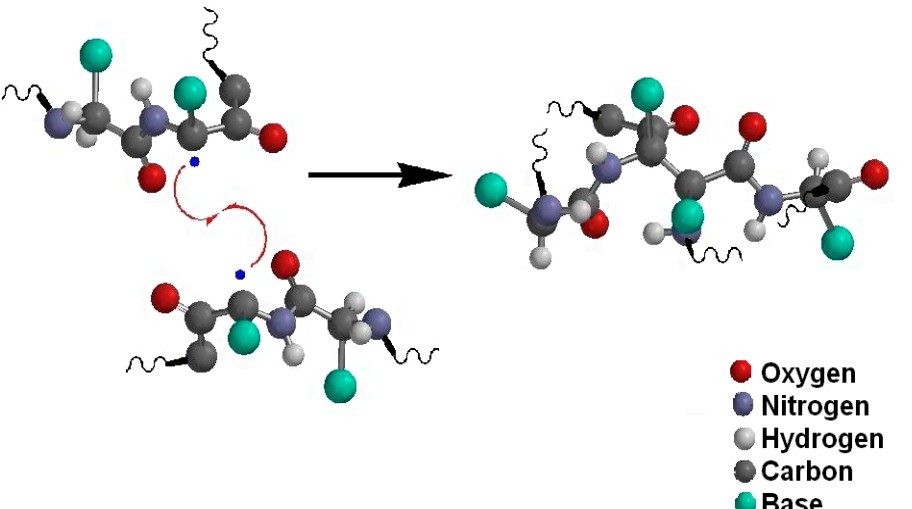

**Figure 4.** The cross-linking of amino acid chains by the reaction of two protein radicals.

## 3. Naturally Occurring Antioxidants

In general, antioxidants refer to a wide range of substances that can be divided into endogenous or exogenous. Oroian and Escriche [36] classified the antioxidants mainly in vitamins (vitamins C and E), carotenoids (carotenes and xanthophylls), and polyphenols (flavonoids, phenolic acids, lignans and stilbenes). On the other hand, Carocho et al. [5] categorized the antioxidants by the most important groups, taking into account their properties, function and applicability in the industry, emphasizing polyphenols and the different groups of carotenoids, within natural antioxidants. Another way to catalog antioxidants is in terms of their solubility, which can be soluble in lipids/fats (hydrophobic) and water (hydrophilic). Both are necessary to protect cells, since the interior of cells and the fluid between is composed of water, while cell membranes are mostly made up of lipids. Because free radicals can attack any part of the cell, both types of antioxidants are needed to ensure the complete protection against oxidative damage [37]. A more precise definition of antioxidants is referring to the substances with the ability to hinder ROS or nitrogen species (RNS), in order to inhibit or stop the propagation of oxidative chain reactions before vital molecules may be damaged [5].

Plenty of the research regarding antioxidants of natural origin is focused on the relationship between human health and the intake of foods containing antioxidants [32,38,39]. Others are related to the extraction of bioactive molecules and their incorporation into food supplements or drugs in order to act in the body more efficiently [36,40]. Antioxidants can also be classified based on the reaction mechanism [41]. For example, antioxidants can neutralize the action of reactive species in cell membranes, through three mechanisms names ad (i) hydrogen atom transfer (HAT), (ii) single electron transfer (SET), and (iii) the ability to chelate metals from transition metal. In this sense, HAT mechanism is based on the ability of an antioxidant (HA) to hinder free radicals such as the peroxyl radical –ROO, by donating a hydrogen atom, stabilizing the peroxyl radical by resonance, according to the Equation (1) [42]:

$$AH + ROO^* \rightarrow ROOH + A^* \tag{1}$$

While the SET mechanism is based on the ability of an antioxidant (HA) to transfer an electron to decrease free radicals, pro-oxidant metals such as $Fe2+$ and $Cu2+$, and carbonyls (Equations (2)–(4)). This mechanism depends on the solvent and pH:

$$ROO^* + AH \rightarrow ROO^* + AH^+ \tag{2}$$

$$AH^+ + H_2O \leftrightarrow A^* + H_3O^+ \tag{3}$$

$$ROO^* + H_3O^+ \leftrightarrow ROOH + H_2O \tag{4}$$

Carotenoids are antioxidants found in foods such as carrots, pomegranates, and grapefruit, while tocopherols are present in oilseeds and green leafy vegetables, and ascorbic acid is found in citrus fruits [43]. Another example of phytochemical antioxidants is flavonoids, which have one of the most diverse groups of compounds present in food. According to the chemical structure, they are divided into six classes which are: flavanols, flavonones, flavones, flavonols, isoflavonoids and anthocyanidins. Because they can cross the blood-brain barrier and reduce oxidative stress in that area, their beneficial health effects include the anti-inflammatory effect, and the prevention and delay of the progression of chronic neurodegenerative diseases [44].

Regarding food preservation, the most studied antioxidants of natural origin are vitamins (tocopherols and ascorbic acid), stilbenes (resveratrol), and polyphenols (gallic acid and quercetin), as well as plant extracts (spices, leaves and fruits) that are a mixture of antioxidant molecules from various families, mainly polyphenols. Table 1 describes the type of antioxidant, description and natural source, applied research in the preservation of food and reference.

**Table 1.** Summary of the major findings of the applied research in the use of antioxidants for the preservation of foods.

| Ref. | Major Findings | Description | Antioxidant |
|------|----------------|-------------|-------------|
| [45] | Aa comparative study between different tocopherols and tocotrienols for the inhibition of the oxidation of vegetable oils and animal fats was carried out. It was found that at low concentrations, $\alpha$-tocopherol is more efficient in scavenging free radicals, while $\gamma$-tocopherol was better at relatively high concentrations. | Tocopherols and tocotrienols | Vitamins |
| [46] | $\alpha$-Tocopherol presented a better antioxidant performance in lipids when it is in the presence of phospholipids such as phosphatidylethanolamine. | | |
| [47] | The synergistic effect between propylgalate and $\alpha$-tocopherol was compared. The authors found that the antioxidant properties in oil-in-water emulsions were greater than when only the tocopherol was used. This effect was attributed to a regeneration of the vitamin by the action of propylgalate. | | |
| [5] | Ascorbic acid not only allows one to maintain the quality of post-harvest vegetables, but also increases their shelf life and improves the properties of vegetables. | Vitamin C or ascorbic acid | |
| [48] | The effect of pre-harvest treatment with ascorbic acid and calcium lactate on bell pepper was studied. They found that the appearance and shelf life of the fruit increased with the treatment, also the amount of flavonoids in the fruit, thereby improving its antioxidant capacity. | | |
| [49] | A decrease in post-harvest enzymatic browning of mango beans of up to 50% when using an ascorbic acid treatment against a control without treatment was reported. They also noted that bean sprouts increased the polyphenol content and antioxidant capacity. | | |
| [50] | The antioxidant properties of resveratrol from the point of view of its chemical structure were studied. It was found that resveratrol inhibited lipid peroxidation by 89% compared to BHT and propylgalate, which had values of 68 and 83%, respectively. | Resveratrol | Stilbenes |
| [51] | Several resveratrol esters with long chain ($C_{14}$, $C_{16}$ and $C_{18}$) and short chain ($C_3$, $C_4$, and $C_6$) fatty acids were prepared and their antioxidant properties with different free radicals compared. It was that the antioxidant properties of long-chain resveratrol esters was better for the 2,2-diphenyl-1-picrylhydrazil (DPPH) radical. On the other hand, short chain esters showed a better antioxidant properties against 2,2′-azino-bis (3-ethylbenzothiazoline-6-sulfonic acid) (ABTS). | | |
| [5] | Recently this antioxidant has received special attention for food preservation, because it imparts an astringent flavor. It is used mainly in acidic juice drinks such as blueberry, and grape juices. Gallic acid esters, such as propylgalate, are used to prevent lipid oxidation. | Gallic acid | Polyphenols |
| [52] | The performance of the synthetic antioxidant tertbutylhydroquinone (TBHQ) was evaluated against gallic acid, and a mixture of methyl gallate with gallic acid, in the thermal oxidation of lipids. During the thermal oxidation, so-called second-stage oxidized species are generated, that is, oxidized products of lipid peroxides. It ws found that at low temperatures TBHQ performs better, but at 120 °C, the gallic acid, and its mixture with methyl gallate, showed a better performance. | | |
| [53] | The antioxidant effect of quercetin, epicatechin and naringenin on methyl linoleate was studied. They found that naringenin had a poor antioxidant effect compared to quercetin and epicatechin. | Quercetin | |
| [54] | The antioxidant properties of quercetin and quercetin with $\alpha$-tocopherol in chicken meat were studied. Greater preservation of the meat was observed under storage conditions when quercetin was used, in addition to the elimination of odors caused by carbonyl compounds. However, the appearance of a yellow color can be avoided if tocopherol is also used in addition to the quercetin. | | |
| [55] | The oxidation of lipids and proteins in chicken pate was evaluated in the presence of quercetin and butylated hydroxytoluene (BHT). Quercetin was found to be eight times more efficient in inhibiting lipid oxidative reactions than BHT. However, quercetin was not as efficient in inhibiting protein oxidation. | | |

## 4. Development of Lipids and Nano-Lipid Vehicles in the Conservation of Antioxidants

Since the early 90s, with the inclusion of antioxidant-rich foods in the daily diet, Artaud-Will et al. [56] demonstrated that these substances could help to reduce cellular oxidative stress [57,58]. Beyond the consumption of antioxidants in the diet or as a supplement, another line of research that has gained relevance is the pharmaceutical use of antioxidants as a possible therapy for different degenerative diseases related to oxidative stress such as cancer, coronary and cerebrovascular diseases [59–62]. However, a challenge faced in this field is the breakdown of antioxidant molecules as they pass through the gastrointestinal tract, which limits their bioaccessibility and bioavailability. To overcome this difficulty, encapsulating materials have been used to protect the bioactive molecules, promoting the absorption in the target organs [63–66]. Tapeinos et al. [67] classified these particles and encapsulating materials used as carriers of bioactive molecules in three general groups based on polymeric, inorganic and lipid particles. Table 2 presents several examples of these types of particles. Although polymeric and inorganic particles can be very efficient in drug transport because of the higher resistance to the gastrointestinal tract, their toxicity is the main disadvantage.

The development of polymeric nanoparticles as carriers for biofunctional molecules was first used in the pharmaceutical industry. However, these advances have been applied also in the preservation of foods. Zambrano-Zaragoza et al. [68] carried out a classification of nanoparticles used as transporters for the preservation of foods, where different organic and inorganic materials are described. In general, the nanocontainers described by these authors are spherical or hemispherical particles (with the exception of haloisite nanotubes and nanofibers) containing bio-functional molecules. In the case of polymeric and solid lipid nanocontainers, these are continuous matrices where smaller particles or aggregates of the bio-functional molecules are dispersed. In self-assembled nanoparticles, the bio-functional molecule is usually located between the lipid molecules forming an external structure or in the liquid phase in the nucleus of the particle. Inorganic materials are structures with cavities where bio-functional molecules are lodged by impregnation. In addition, core-shell structures are covered with a third component to improve stability to environmental factors. However, nanostructures obtained from polymers or inorganic materials may cause reactions with detriment on health or even cause cellular damage. Therefore, nanostructures based on materials of natural origin such as liposomes, solid lipid nanoparticles (SLNs) and nanostructured lipid carriers (NLCs) arise as excellent candidates in the preservation of foods [67,69,70]. SLNs are colloidal particles with sizes in the range of 50–1000 nm, built of lipids with a relatively high melting point. These are transport agents with good biocompatibility and low toxicity for functional molecules such as drugs, conceived with the aim of preventing their decomposition and providing stability [71]. In SLNs, the bioactive molecule is stored in the continuous solid lipid matrix. However, three different arrangements are developed according to the way the bioactive molecules are dispersed within the solid lipid matrix. (i) In the bioactive-enriched matrix (BEM), the bioactive molecule is homogeneously dispersed throughout the lipid matrix. (ii) In the bioactive-enriched shell (BES), the bioactive molecule is concentrated near the surface of the particle giving the appearance of forming a shell of active molecules. (iii) In the bioactive-enriched nucleus (BEN), the bioactive molecule is concentrated in the center of the particle, forming a well-established nucleus. Figure 5 schematizes the three types of SLNs. Suspension methodologies for synthesizing these nanoparticles include high shear stress techniques such as high-pressure homogenization (HPH), and ultrasonic radiation, or low shear techniques such as microemulsion, membrane filtration, and temperature phase inversion [72].

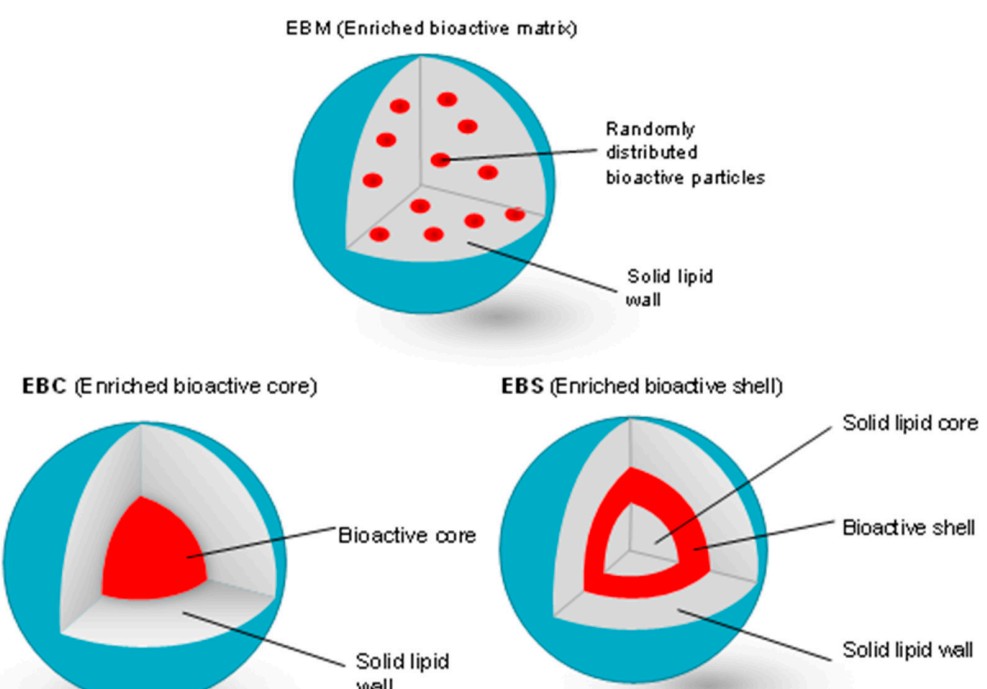

**Figure 5.** Three types of SLN. Enriched bioactive matrix (EBM), enriched bioactive core (EBC), and enriched bioactive shell (EBS).

Jose et al. [73] encapsulated resveratrol in SLNs to increase its stability until reaching the target tissue. Oehlke et al. [74] loaded SLNs with tocopherol and ferulic acid, and stored the samples for 6 and 15 weeks. After these periods, the antioxidant capacity of the samples was determined and compared. The tocopherol-loaded particles increased the antioxidant activity, while the ferulic acid-loaded particles maintained the stable antioxidant capacity. The behavior of tocopherol was attributed to the crystallization of the lipid, which caused the tocopherol to be expelled towards the surface of the particle, being more exposed and available for the neutralization of free radicals. Jain et al. [75] evaluated the antioxidant properties of β-carotene encapsulated in SLNs. According to them, after 3 months of storage, the antioxidant activity of encapsulated β-carotene was higher than that without encapsulation. One of the disadvantages of SLNs is that lipid crystallization can expel the biofunctional molecule during storage time. In addition, high-energy preparation methods can decompose thermolabile molecules, while low-energy ones, such as emulsions, require a high concentration of surfactant that is not desirable in a food product [76]. Due to these disadvantages, SLNs are not widely studied in food preservation applications.

## 5. Application of Antioxidants in Edible Films

The use of barriers in food is an efficient way to reduce moisture loss, limit gas exchange and reduce the damage caused by oxidizing agents and microorganisms. Packaging normally fulfills this function in processed foods, but in the case of fresh food, plastic packaging is not always the best alternative, since it does not regulate the exchange of vapors, avoids the passage of light nor inhibits the deterioration caused by microorganisms or enzymes of live cells [3,77,78]. One strategy to improve the performance of conventional package, is the incorporation of functional molecules or particles that can induce properties such as antioxidant, antimicrobial, while act as selective gas barrier [2,3,79]. Thus, the use of edible films (EFs) has become an important alternative in the food industry to protect fresh, minimally processed foods and even those subjected to a cooking process [2,79]. There is extensive research on EFs functionalized with antioxidant substances, which may include plant extracts, essential oils, or isolated and purified compounds [1]. For example, Sabaghi et al. [80] evaluated the sensory, antioxidant and fungistatic properties of a coating

made with chitosan and green tea extract plasticized with glycerol applied to walnut hearts, which remained in storage conditions at room temperature for 18 weeks. They concluded that fungal growth was not detected in the coated walnut core samples regardless of the concentration of green tea extract. However, oxidation of walnut lipids was inversely correlated with the concentration of tea extract. The higher concentration of extract led to an alteration of the sensory properties, considered unacceptable by the evaluating panelists. Bermudez et al. [81] studied the antioxidant effect of coatings based on pectin, gelatin, and beeswax, and plasticized with glycerol and sorbitol. The mixtures for the coating were functionalized with two antioxidants present in the olives, hydroxytyrosol and 3,4-dihydroxyphenylglycol. The coating was applied on beef stored for 7 days at 4 °C. The determination of reactive substances for thiobarbituric acid (TBARS) indicated that the samples coated with the films incorporated with antioxidants, significantly inhibited the oxidation processes of the meat compared to the control samples without antioxidants. Xiong et al. [82] prepared a coating based on chitosan and gelatin, functionalized with two different antioxidants, grape seed extract, and nisin. The mixtures were applied as a coating on fresh pork meat under storage conditions of 4 °C for 20 days. The deterioration of the meat was measured with different parameters such as color change, lipid oxidation measurement, protein oxidation measurement and microbiological viability. They found that the coating based on chitosan, gelatin and grape seed extract showed a better performance in the conservation of the meat with respect to the uncoated control samples or coated samples without antioxidants. In addition, nisin-functionalized samples showed no antioxidant or antimicrobial effect. Tongdeesoontorn et al. [83] reported a coating mixture of starch with carboxymethyl cellulose, functionalized with two quercetin and tertbutylhydroquinone (TBHQ). The antioxidant properties of the prepared films were measured by the reaction with 2,2-diphenyl-1-picrylhydrazyl radicals (DPPH) to evaluate the antioxidant effect in the oxidation process of lard and pork, while the degree of oxidation in the butter was evaluated by means of the peroxide index. One of the parameters to evaluate the meat oxidation process was the color change. Neither of the two antioxidants showed a significant difference in terms of radical scavenging activity with respect to the film without antioxidants, this was attributed to the content of polyphenols present in the starch. However, both quercetin and TBHQ functionalized films showed a considerable decrease in peroxide value in lard and loss of color in meat when compared to the control sample. Table 2 lists different EFs registered products, the overall composition, and the field of application.

In addition, functional films based on bio-materials can also be used to modify food packaging materials. Wu et al. [84] deposited on polyethylene, chitosan liposomes loaded with laurel essential oil (LEO) and silver nanoparticles. Films with liposomes presented better antimicrobial and antioxidant properties, since they decreased the oxidation of lipids in pork, as well as retarded bacterial growth, prolonging the shelf life of the meat stored at 4 °C and wrapped in the protective film, from 9 days to 14 days, for uncoated and functionally coated films, respectively. Esmaeili et al. [85] incorporated garlic essential oil (GEO)-loaded liposomes into edible films chitosan and WPI. The films of both polymers improved their antioxidant and antimicrobial properties, compared to films that did not contain GEO and to films with unencapsulated GEO. In either case, the chitosan films performed better than the WPI films. Giteru et al. [86] indicated that polymers such as proteins, polysaccharides and lipids have been used in the production of edible films, because of their biodegradability and compatibility with foods. Likewise, incorporating bioactive antimicrobial and antioxidant compounds contributes to increasing the shelf life of foods. They reported the effect of incorporating citral and quercetin as bioactive compounds in Kafirin films, to prolong the shelf life of chicken fillets during cold storage. From the four elaborated systems, they found that citral kafirin (CK) and kafirin-citral-quercetin (CKQ) films showed greater mechanical flexibility and a decrease in microbial proliferation than in those chicken fillets with kaferin-quercetin (KQ) films. Farrag et al. [87] characterized the release kinetics of quercetin-loaded starch films of legume and cereal

origins in donut-shaped structures. They reported that the antioxidant activity was higher in the films made from leguminous starch than those obtained from cereal starch. In vitro release from the quercetin-loaded films was carried out in an aqueous-ethanolic medium. The released quercetin reached the equilibrium in 1−4 days for the cereal starch films, and in more than 7 days for the films of legume origin. Jamróz et al. [88] made biodegradable films of furcellaran and gelatin (FUR/GEL) and incorporated extracts of pu-erh (RTE) and green tea (GTE). They examined the physicochemical properties (thickness, density, solubility in water, moisture, color and degree of swelling), antioxidants (DPPH, and ABTS radical scavenging), antimicrobial (*Escherichia coli*, *Staphylcoccus aureus*, *Candidia albicans* and *Henseniaspora uvarum*), and mechanical (tensile strength and breakage). They reported FUT/GEL-RTE and FUT/GEL-GTE systems at concentrations of 5, 10, and 20% of each extract. The incorporation of GTE or RTE in the films significantly reduced water solubility and the improved the mechanical behavior. The GTE films changed their color during the spoilage test on fish, indicating that they could be used as smart pH indicators. In relation to the different concentrations of extracts used, they found that 20% showed the highest antimicrobial and antioxidant activity. This concentration was selected for the packaging of food products.

**Table 2.** Some edible films products available in the market [89].

| Tradename | Manufacturer or Brand | Composition | Uses and Advantages |
|---|---|---|---|
| **Crystalac Z2**® | Mantrose-Haeuser Co., Inc. | Zeín | Jams and glazes |
| **SemperFresh**® | Pace International | Short chain sucrose esters, fatty acids and sodium salts of carboxymethyl cellulose | Coating for cherries. A Selective barrier of humidity and gases, avoids weight losses due to dehydration, preservation of color |
| **Nita**® | Nita Casings | Collagen or alginate | Casing or forming packaging for the protection of meat sausages during and after cooking. Selective gas barrier |
| **Pro-Long**® | Tal Chemicals Co. | Sucrose, fatty acid polyesters ansodium salts of carboxymethyl cellulose | Coating for freshly cut fruits and vegetables |
| **NatureSeal**® | AgriCoat/NatureSeal | Hydroxypropylmethyl cellulose with ascorbic acid and calcium chloride | Inhibits browning and maintains the texture and flavor of freshly cut fruits and vegetables |

## 6. Conclusions

This review was focused on the use of edible films enriched with liposomes loaded with antioxidants, as an alternative to prolong the shelf life of food products. The oxidation mechanisms of proteins and lipids, which are the main cause of the deterioration of food during storage, were described. The antioxidant reaction mechanism and its application in food preservation were presented. Likewise, the action of liposomes loaded with antioxidants was described and some examples of their application in edible films were shown. From the reviewed bibliography, it is possible to understand that this field is still in its initial stages, with a very promising future in the commercial and industrial sectors.

**Author Contributions:** Conceptualization, M.R.-A. and M.Z.S.-L.; investigation, M.R.-A. and M.Z.S.-L.; data curation, C.L.-P. and P.C.-A.; writing—original draft preparation, M.Z.S.-L. and C.L.-P.; writing—review and editing, M.Z.S.-L. and C.L.-P.; visualization, J.T.P.-U. and M.Z.S.-L.; supervision, P.C.-A. and C.L.-P. All authors have read and agreed to the published version of the manuscript.

**Funding:** This research received no external funding.

**Institutional Review Board Statement:** Not applicable.

**Informed Consent Statement:** Not applicable.



**Data Availability Statement:** Not applicable.

**Acknowledgments:** M.R.-A. is grateful to the Consejo Nacional de Ciencia y Tecnología (CONACYT) in Mexico for the financial support provided during his Ph.D. studies through the scholarship No. 2018-000068-02NACF-24366. C.L.-P. thanks to Paola Balderrama by the English corrections in final version of the manuscript.

**Conflicts of Interest:** The authors declare no conflict of interest.

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
