# Peer review of "Application of Antioxidants as an Alternative Improving of Shelf Life in Foods"

_2673-4176, doi:10.3390/polysaccharides2030036_

Round 1

Reviewer 1 Report

The manuscript "Application of antioxidants as an alternative in the improving of shelf life in foods" interestingly introduces the topic of the use of natural substances in extending the shelf life of food.

Detailed comments:

The manuscript is very well written and presents the most important discoveries in this topic.

The prepared Figure is nice and legible.

Some lines are repeated in Table 1.

The literature for the manuscript was very well chosen. Articles from 2013-2021 were mainly presented, which proves the significant development of the topic in recent years. 

Author Response

RESPONSE TO REVIEWERS

Dear reviewers, the authors appreciate the suggestions, observations and comments made in favor of improving the manuscript. They were all carried out doing our best. The changes can be viewed in the word file named "track changes".

Simple corrections were made to the English language.

In general, reviewers 1 and 2 agreed to add more references in some sentences. In total, 21 new references were added in different parts of the text.

Table 1 was corrected and rearranged, as indicated by reviewers 1 and 2.

Reviewers 2 and 3 suggested rearranging the keywords. This was done as suggested by reviewer 2 (i.e. polysaccharides; antioxidants; loaded liposomes; edible films; food conservation).

Reviewers 2 and 3, suggested highlighting the contribution of the work. This was done at the end of the Introduction as:

“The contribution of the work lies in the use of polysaccharides as carrier agents in the development of antioxidants loaded liposomes and their incorporation in edible films.”

Reviewer 3 suggested changing the order of the topics. However, the authors consider that the order presented has a "systematic order". That is, the manuscript starts with the topic of antioxidants, continue with the development of antioxidant-loaded liposomes, and end with the incorporation of these liposomes in edible films.

Reviewer 3 suggested to improve the quality of figure 5. Although at simple sight it was no easily observed, figure was slightly modified to contrast the differences in the three types of solid lipid nanoparticles.

Reviewer 2 Report

General comments

The submitted manuscript “Application of antioxidants as an alternative in the improving of shelf life in foods” consists in a mini-review focused on the use of antioxidant-enriched edible films.

The topic of review is interesting and well matches the aim and scope of Polysaccharides.

The Tables have to be reorganized in order to make them clearer and more incisive. Further Figures and schematizations should be added.

In addition a moderate revision of the English language.

Some suggestions are reported below point by point.

Keywords

The chosen keywords (i.e. antioxidants; polysaccharides liposomes; food conservation; edible films) should be reported in a more logical order (materials, processing, characterisation, properties, applications).

  1. Introduction

- The following period “Natural occurring antioxidants are classified according to the chemical structure in carotenoids, vitamins, polyphenols, quinones, and minerals. Unfortunately, due to pH conditions in the stomach, or exposure to atmospheric air, natural antioxidants break down quickly, losing their properties and limiting the antioxidant activity. One strategy to prolong its effect is through the microencapsulation with carrying agents. These agents are based on carbohydrate polymers, which in turn are made up of polysaccharide chains such as glucose and fructose” needs to be supported with suitable references, including “Strategies To Improve Ellagic Acid Bioavailability: From Natural Or Semisynthetic Derivatives To Nanotechnological Approaches Based On Innovative Carriers,  Nanotechnology 31[38] (2020): 382001”.

Moreover, in general, it is not true that the carrying agents are based on carbohydrate polymers…there are many other kinds, such as polyesters. This point has to be clarified, adding proper references, such as “Controlled release of 18-β-glycyrrhetic acid by nanodelivery systems increases cytotoxicity on oral carcinoma cell line, Nanotechnology 29[28] (2018) 285101 (11pp).”

- The following considerations “Although packaging is an alternative to protect the quality of food and improve its shelf life, delaying microbial deterioration and providing barrier properties against moisture and gases, this alternative is not environmentally friendly. Due to the above, edible films made from natural polymers such as polysaccharides, proteins and fats are a potential alternative to solve this problem with a lower impact on the environment” are not clear, since also food packaging could be fabricated in sustainable materials and could be characterised by the antioxidants addition. Please clarify this point, adding proper references.

- The review paper aim is clearly reported, but the originality is not clear and has to be better evidenced, at the end of the Introduction section.

  1. Naturally occurring antioxidants

More details about natural antioxidants can be added, since many interesting ones have been omitted

  1. Development of lipids and nano-lipid vehicles in the conservation of antioxidants
    - The following sentences “However, a challenge faced in this field is the breakdown of antioxidant molecules as they pass through the gastrointestinal tract, which limits their bioaccessibility and bioavailability. To overcome this difficulty, encapsulating materials has been used to protect the bioactive molecules, promoting the absorption in the target organs.” have to be supported with proper references.

- The following statement “Although polymeric and inorganic particles can be very efficient in drug transport because of the higher resistance to the gastrointestinal tract, the little biocompatibility is the main disadvantage” is not true…it depends on the selected polymers and inorganic particles. The Authors should better expand this concept and support with appropriate references.

- The following statement “The development of these advances in the pharmaceutical industry has also been applied in the preservation of foods” needs proper references.

  1. Application of antioxidants in edible films

As a general consideration, in this paragraph not only edible films are described but also coatings on packaging systems. It has to be specified.

- The following sentences “The use of barriers in food is an efficient way to reduce moisture loss, limit gas exchange and reduce the damage caused by oxidizing agents and microorganisms. Packaging normally fulfills this function in processed foods, but in the case of fresh food, plastic packaging is not always the best alternative, since it does not regulate the exchange of vapors, avoids the passage of light or inhibits the deterioration caused by microorganisms or enzymes of live cells” have to be corroborated with proper references.

- Moreover, it has to be specified which kinds of barrier can be used, such as the use of inorganic fillers, citing related references, such as “Eco-sustainable systems based on poly(lactic acid), diatomite and coffee grounds extract for food packaging, Intern J Biolog Macromolecules 112(2018): 567-575.”.

Tables

  • In all Tables the outcomes should be reported in a shorter and more schematic manner.

Author Response

RESPUESTA A LOS REVISORES

Estimados revisores, los autores agradecen las sugerencias, observaciones y comentarios realizados a favor de la mejora del manuscrito. Todos se llevaron a cabo haciendo nuestro mejor esfuerzo. Los cambios se pueden ver en el archivo de Word llamado "seguimiento de cambios".

Se hicieron correcciones simples al idioma inglés.

In general, reviewers 1 and 2 agreed to add more references in some sentences. In total, 21 new references were added in different parts of the text.

Table 1 was corrected and rearranged, as indicated by reviewers 1 and 2.

Reviewers 2 and 3 suggested rearranging the keywords. This was done as suggested by reviewer 2 (i.e. polysaccharides; antioxidants; loaded liposomes; edible films; food conservation).

Reviewers 2 and 3, suggested highlighting the contribution of the work. This was done at the end of the Introduction as:

“The contribution of the work lies in the use of polysaccharides as carrier agents in the development of antioxidants loaded liposomes and their incorporation in edible films.”

El revisor 3 sugirió cambiar el orden de los temas. Sin embargo, los autores consideran que el orden presentado tiene un "orden sistemático". Es decir, el manuscrito comienza con el tema de los antioxidantes, continúa con el desarrollo de liposomas cargados de antioxidantes y termina con la incorporación de estos liposomas en películas comestibles.

El revisor 3 sugirió mejorar la calidad de la figura 5. Aunque a simple vista no se observó fácilmente, la figura se modificó ligeramente para contrastar las diferencias en los tres tipos de nanopartículas sólidas de lípidos.

Reviewer 3 Report

Manuscript Number: polysaccharides-1261197-peer-review-v1

Title: Application of antioxidants as an alternative in the improving of shelf life in foods.

Certainly, the authors have made an important effort to writing the article. There are some comments with this script. Few of them are mention below there for author attention

  1. Modify the key words to make it more consistent with the manuscript.
  2. Check and correct the wrong order of the chapters.
  3. Some of the figures in the manuscript, such as Figure 5, are not clear enough. Please adjust them.
  4. What is the originality in this work, please highlight it.

Author Response

(The authors gave the same response as above.)

Round 2

Reviewer 2 Report

General comments

The Authors should have replied not in general to all Reviewers (and only using English language), but specifically to each one. They did not replied to all requests, as reported below.

  1. Introduction

- As already evidenced, it is not true that the carrying agents are based on carbohydrate polymers…there are many other kinds, such as polyesters. This point has to be clarified, adding proper references, such as “Controlled release of 18-β-glycyrrhetic acid by nanodelivery systems increases cytotoxicity on oral carcinoma cell line, Nanotechnology 29[28] (2018) 285101 (11pp).”

  1. Naturally occurring antioxidants

More details about natural antioxidants can be added, since many interesting ones have been omitted

  1. Application of antioxidants in edible films

As a general consideration, in this paragraph not only edible films are described but also coatings on packaging systems. It has to be specified in the title.